# Search for Novel Potent Inhibitors of the SARS-CoV-2 Papain-like Enzyme: A Computational Biochemistry Approach

**DOI:** 10.3390/ph15080986

**Published:** 2022-08-11

**Authors:** Manuel I. Osorio, Osvaldo Yáñez, Mauricio Gallardo, Matías Zuñiga-Bustos, Jorge Mulia-Rodríguez, Roberto López-Rendón, Olimpo García-Beltrán, Fernando González-Nilo, José M. Pérez-Donoso

**Affiliations:** 1Center for Bioinformatics and Integrative Biology (CBIB), Facultad de Ciencias de la Vida, Universidad Andres Bello, Av. República 330, Santiago 8370146, Chile; 2Facultad de Medicina, Centro de Investigación Biomédica, Universidad Diego Portales, Ejército 141, Santiago 8320000, Chile; 3Center of New Drugs for Hypertension (CENDHY), Santiago 8380494, Chile; 4Núcleo de Investigación en Data Science, Facultad de Ingeniería y Negocios, Universidad de las Américas, Santiago 7500000, Chile; 5Center for Bioinformatics, Simulation and Modelling, Facultad de Ingeniería, Universidad de Talca, Talca 3465548, Chile; 6Laboratorio de Bioingeniería Molecular a Multiescala, Facultad de Ciencias, Av. Instituto Literario 100, Universidad Autónoma del Estado de México, Toluca 50000, Mexico; 7Facultad de Ciencias Naturales y Matemáticas, Universidad de Ibagué, Carrera 22 Calle 67, Ibagué 730002, Colombia; 8Centro Integrativo de Biología y Química Aplicada (CIBQA), Universidad Bernardo O′ Higgins, General Gana 1702, Santiago 8370854, Chile

**Keywords:** papain-like protease of SARS-CoV-2, molecular dynamics simulation, binding free energy, virtual screening

## Abstract

The rapid emergence and spread of new variants of coronavirus type 2, as well as the emergence of zoonotic viruses, highlights the need for methodologies that contribute to the search for new pharmacological treatments. In the present work, we searched for new SARS-CoV-2 papain-like protease inhibitors in the PubChem database, which has more than 100 million compounds. Based on the ligand efficacy index obtained by molecular docking, 500 compounds with higher affinity than another experimentally tested inhibitor were selected. Finally, the seven compounds with ADME parameters within the acceptable range for such a drug were selected. Next, molecular dynamics simulation studies at 200 ns, ΔG calculations using molecular mechanics with generalized Born and surface solvation, and quantum mechanical calculations were performed with the selected compounds. Using this in silico protocol, seven papain-like protease inhibitors are proposed: three compounds with similar free energy (D28, D04, and D59) and three compounds with higher binding free energy (D60, D99, and D06) than the experimentally tested inhibitor, plus one compound (D24) that could bind to the ubiquitin-binding region and reduce the effect on the host immune system. The proposed compounds could be used in in vitro assays, and the described protocol could be used for smart drug design.

## 1. Introduction

Since the first outbreak in the city of Wuhan in 2019, the type 2 coronavirus (CoV-2) that causes severe acute respiratory syndrome (SARS) has kept health systems on alert [1]; SARS-CoV-2 has caused ~500 billion infections and ~6 million deaths worldwide to date, in addition to significant economic losses. The appearance of more contagious viral variants with the ability to evade the immunity achieved by the vaccine has reinforced the necessity to search for new treatments [2]. The spread of some dangerous variants of the virus is alarming, such as δ, which was first detected in India and is currently present in more than 60 countries [3]. This variant presents mutations that decrease the effectiveness of the antibodies, whether they are monoclonal, generated by a previous SARS-CoV-2 infection or those obtained by the vaccine against the virus [4,5]. Faced with the rapid evolution of the virus and the possibility of future outbreaks, it is necessary to develop new treatments. For this reason, the search for antiviral compounds is essential to treat patients with the most severe forms of the disease, those that do not tolerate vaccines or that were infected with more dangerous variants of the virus.

Based on the molecular mechanisms determining the progression of the viral infection, it is possible to propose enzyme inhibitors that could become effective pharmacological therapies against SARS-CoV-2 [6]. After entry into the cell mediated by Angiotensin Converting Enzyme 2 (ACE-2), SARS-CoV-2 loses its envelope and synthesizes polyproteins 1a and 1ab (PP1a and PP1ab) [7]. These polypeptides contain immature forms of the proteins that will be part of the virion (structural), and that will participate in the maturation of the viral particles (nonstructural) [8]. The action of viral proteases on PP1a and PP1ab releases the mature structural and nonstructural proteins, which initiate the synthesis of the components of the new viral progeny. This is one of the fundamental stages for the replication of SARS-CoV-2 in such a way that its inhibition stops viral replication. Based on this background, currently, the scientific and pharmaceutical community has devoted great efforts to searching for drugs that block the protease activity of SARS-CoV-2 [6].

There are two viral proteases, the “major” 3-chymotrypsin-like protease (3CLpro or Main protease) and the papain-like protease (PLpro) [7]. Main protease (Mpro) is a cysteine protease with different cleavage sites than human proteases, making it an interesting target for developing inhibitors [9,10]. PLpro, in addition to protease activity, presents deubiquitinase and deISGylase (the ability to deconjugate interferon-stimulated gene 15 protein, ISG15, from substrates), which affects the immune response during infection [11,12]. Thus, molecules with the ability to inhibit PLpro could also inhibit viral replication and reduce the impact of the virus on the host’s immune system.

PLpro has several domains with independent activity. In the N-terminus region, it presents a Ubiquitin-like domain (Ubl2) whose activity could affect the sensitivity to interferon (IFN), probably mediated by the deubiquitination of the nuclear factor Kappa-light-chain-enhancer of B cell (NFk-b) [13]. The PLpro protease domain recognizes the Lys-X-Gly-Gly sequence by cleaving the PP1a and PP1ab polyprotein to initiate virus maturation [8]. Furthermore, as has been observed through in vitro studies, it has ubiquitinase (Ub) and ubiquitin-like protease activity of gene 15 stimulated by interferon (ISG15), which affects the innate immune response [12,13]. In addition, a Zn finger domain can be identified located between the domains with PLpro and Ubl2 activity, which could participate in the interaction with the different substrates [14,15]. According to its multiple activities, PLpro could be an excellent target for the design of antiviral compounds, as has already been proven by in vitro experiments.

PLpro inhibitors have been developed by computational and experimental studies. However, these compounds have shown weak inhibitory activity in the order of µM. Among the proposed drugs, those designed by adding chemical groups to 5-Amino-2-Methyl-N-[(1r)-1-Naphthalen-1-Ylethyl]benzamide (GRL0617) [16] stand out. Using this strategy, it was possible to obtain a fluorinated molecule called C19, with an IC50 of ~0.44 µM for protease and deubiquitinase activity [17]. Considering an analogous strategy (Figure 1), the PubChem database was scanned for compounds structurally similar to GRL0617 (Tanimoto index 80% [18]). By molecular docking with ~18,000 compounds obtained in the search, 500 compounds with higher affinity than GRL0617 for PLpro were selected. Finally, the seven compounds with the highest ligand efficacy index and ADEM-Tox properties within acceptable ranges for drugs were selected. With these selected compounds, molecular dynamics simulation studies were performed to calculate the binding free energy (ΔG_bind_) by MMGBSA [19,20,21] and quantum mechanical calculations and chemically characterize the protein/ligand interaction. MMGBSA calculations decompose ΔG_bind_ into a solvation term (ΔG_solv_) and a gas-phase term (ΔG_gas_) [22]. ΔG_gas_ includes the van der Waals energies (ΔE_vdw_) and the electrostatic contribution (ΔE_ele_), while ΔG_solv_ includes the energies associated with the polar (ΔE_GB_) and nonpolar (ΔE_SA_) terms. From these analyses, we identified compounds D28, D04, and D59 with ΔG_bind_ like that calculated for PLpro/GRL0617, and molecules D60, D99, and D06 with ΔG_bind_ more negative than that calculated for PLpro/GRL0617. Furthermore, as observed in molecular dynamics simulation, compound D24 can bind to the ubiquitin-binding region and may inhibit the ubiquitinase of the enzyme. The present study proposes PLpro inhibitors to initiate experimental studies that could establish their inhibitory capacity for use in future pharmacological treatments.

## 2. Results and Discussion

GRL0617 has been reported to inhibit the protease activity of PLpro from SARS-CoV-2, and crystallographic studies have determined the structural details supporting its activity [11]. Considering this promising background, other researchers have studied the activity of other compounds with chemical or structural similarity to GRL0617, obtaining candidates with higher affinity for PLpro [17]. However, groups of less than 500 thousand compounds have been explored, and the selection has been made mainly based on physicochemical descriptors. In the present work, we performed a search for compounds structurally similar (Tanimoto Index 85%) to GRL0617 in the PubChem database containing ~109 million molecules.

### 2.1. Molecular Docking

To evaluate PLpro/ligand binding, molecular docking was used, and the most stable conformation and ΔG_bind_ of the protein in complex with 17,889 compounds obtained from the first screening were determined. In addition, the same calculations were performed for the PLpro/GRL0617 (−10.0 kcal mol^−1^) and PLpro/12C (−8.8 kcal mol^−1^) complexes, two compounds with proven in vitro activity and whose crystal structures have been published. GRL0617 is one of the first specific inhibitors of PLpro, and 12C was designed by incorporating an N-phenylacetamide group into the amide group of GRL0617 [17]. Despite the large conformational space accessible to the protein/ligand complexes, the most stable conformations obtained by docking for PLpro/GRL0617 and PLpro/12C show a high similarity to their crystallographic structures. The structure of the best PLpro/GRL0617 docking is similar to the published (7CMD) crystal (RMSD of 2.4 Å) (Appendix A). In the case of compound 12C, unlike what was observed with GRL0617, the PLpro/12C coupling calculation generates a different structure (Appendix A) than the published (7E35), but at the same site and with a considerable ΔG_bind_.

The ΔG_bind_ energy of the PLpro/GRL0617 complex (−10.0 kcal mol^−1^) was used as a threshold, and 500 compounds with a ΔG_bind_ between −11 and −14.1 kcal/mol were selected (Appendix A). In addition, the crystal structures of PLpro/GRL0617 and PLpro/C12 were used as controls allowing us a comparative analysis for each stage of the selection, mainly those associated with free energy calculations and MD simulation.

### 2.2. Physicochemical Descriptors and ADME Properties

The 500 compounds with structural similarity to GRL0617 and with the highest ΔG_bind_ were sorted according to descending LE to select the 20 with the highest value of this physicochemical descriptor. To assess the drug viability of each compound, six physicochemical properties were calculated: lipophilicity, size, polarity, solubility, flexibility, and saturation. After discarding compounds that did not meet more than two pharmacological criteria, the seven with the highest LE were selected. As shown in Figure 2, compounds D24, D28, D04, D59, D60, D60, D99, D06, 12C, and GRL0617 show up to two parameters out of range. Like GRL0617, which exceeds one ADEM-Tox property (unsaturation), three selected compounds exceed one parameter: compound D06 exceeds the flexibility range and compounds D99 and D60 exceed the flexibility and unsaturation parameters. As shown in Appendix A, all compounds, including the controls GRL0617 and 12C, show a high probability of gastrointestinal absorption, while the controls also show a high probability of brain penetration. Looking at the chemical structures of these compounds (Appendix A), it is identified that molecules D24, D28, D59, D60, D99, and D06 present a naphthyl group, as do GRL0617 and 12C, while compound D04 presents a (9,10-dioxoanthracene-2-yl) amino]-1-oxopropan-2-yl group. In this theoretical framework, the selected compounds could be drug candidates. However, it is necessary to evaluate their affinity for PLpro using a methodology with higher predictive power. In this aspect, MD simulation allows the evaluation of the protein–ligand interaction with a temporal and spatial resolution that is not possible to obtain so far through experimental studies.

### 2.3. Molecular Dynamics Simulation and Protein/Ligand Interactions

From the structures obtained by molecular docking, 200 ns MD simulations were performed in a TIP3P explicit water-box model. These simulations show a stable (Appendix A) PLpro/ligand interaction during the whole simulation time (Figure 3A,B,D–I), except for the PLpro/D24 model that dissociates and binds to another region of the protein (Figure 3C). The simulation time is sufficient to discriminate between ligands that can bind specifically to the GRL0617 binding site and those that cannot (D24). However, for the case of D24, with a 400 ns simulation, we could observe a binding site distinct from that of GRL0617, and that was maintained for more than 200 ns. The stability shown could indicate a new binding region that could be explored in future work. The protein–ligand interaction is visualized in Figure 4A–I, highlighting the residues that maintain proximity of less than 3 Å for more than 80% of the simulation (red and orange). To identify residues that could participate in ligand stabilization, amino acids that remained within 3 Å (close residues) of the ligand for more than 80% of the simulation were identified (Figure 4D–I). For compounds GRL0617 (Figure 4A), 12C (Figure 4B), D28 (Figure 4D), D04 (Figure 4F), D60 (Figure 4G), D99 (Figure 4H), and D06 (Figure 4I), the nearby residues are Leu165, Asp167, Tyr267, and Gln272. Compounds D04, D59, D60, GRL0617, and 12C present the same nearby residues, which include Pro251, Tyr271, and Tyr276, in addition to those already mentioned. For the case of D24, the interaction with the protein is weaker, maintaining interactions in less than 40% of the simulation.

The ligand generates a local perturbation in the protein that can be observed by means of the root mean square fluctuation (RMSF). Figure 4I shows that compounds 12C (green line) and D28 (yellow line) generate higher fluctuation of the polypeptide chain compared to the protein without a ligand (black dashed line). There are four regions of major perturbation in the protein, one near the N-terminus (residues 15 to 60), and two near the binding site (135 to 175 and 180 to 240), which includes the Zinc finger domain and the compound binding region (255 to 300). To identify amino acids that can interact with the compounds, simulations were studied with the CPPTRAJ program Hbond, which uses geometric criteria to determine hydrogen bonds (Figure 5). From the analysis, it is observed that Asp 166 could act as a hydrogen acceptor for compounds D28 (Figure 5D), D04 (Figure 5F), D60 (Figure 5G), D99 (Figure 5H), and D06 (Figure 5I), in addition to the control GRL0617 (Figure 5A). Other residues of possible significance are: Tyr270 for GRL0617 (Figure 5A); Gly165 and Gln271 for compound 12C (Figure 5B); Thr77 and His75 for D24 (Figure 5C); Tyr266, Tyr270, Gln271, and Tyr275 for D28 (Figure 5D); Tyr275 for D04 (Figure 5E); Tyr270 and Tyr275 for D59 (Figure 5F); Thr303 and Asp304 for D60 (Figure 5G); and Asp304 for D06 (Figure 5I).

### 2.4. Non-Covalent Interactions

To obtain a representative conformation of the ligand/protein complex, a cluster analysis of 200 ns of the simulation was performed with the DBSCAN algorithm. From the analysis, structures representing 84% of the simulation for the PLpro/12C complex and more than 91% of the simulation for the other complexes were obtained (Appendix A). These representative structures show the most relevant interactions for the stabilization of the PLpro/ligand complex. To characterize these interactions, NCI calculations were performed (Figure 6), showing by means of an electron density profile, repulsive interactions in red, weak interactions in green, and hydrogen bonds (HB) in blue. As can be seen, the NCI calculations show that weak (van der Waals) interactions between ligand and protein prevail, although some well localized stronger interactions can be identified. As seen in Figure 6A, D–I, ligands can interact with the protein via HB. The nitrogen attached to the benzyl group of GRL6017 can form an HB with hydrogen from the amide group of Gln272, and the carbonyl oxygen of this ligand can act as an acceptor for the hydroxyl proton of the side chain of Tyr267 (Figure 6A). Two carbonyl oxygens of compound D28 can act as a hydrogen acceptor of the peptide bond of residue Glu164 (Figure 6D). Asp167 can act as a proton acceptor forming an HB with an NH close to the cycloheptane of compound D04 (Figure 6E). An HB can be established between the hydrogen of the peptide bond of residue Gln272 and carbonyl oxygen of compound D59 (Figure 6F). Residues Asp167, Tyr276, and Thr304 can act as proton acceptors to form HB with amino groups of compounds D60; in addition, the peptide bond of residue Gln272 can act as a proton donor to form an HB with the carbonyl oxygen of this compound. The side chains of residues Asp167 and Tyr276 can form an HB of compound D99 acting as proton acceptors. An amino group of compound D06 can act as a proton donor to form an HB with the side chain of residue Asp167 and a carboxyl group as a proton acceptor to form an HB with the peptide bond of Gln272.

### 2.5. Free Energy of Binding by MMGBSA

It has been reported in vitro that compounds GRL0617 and 12C effectively inhibit PLpro protease (IC50 ~7 µM) and ubiquitinase (IC50 ~2 µM) activities. Based on this background, we performed MMGBSA calculations to estimate the ΔG_bind_ of the PLpro/ligand complex for each selected compound and of the PLpro/GRL0617 and PLpro/12C complexes (Table 1). The ΔG_bind_ allows the estimation of the affinity of a compound for a target protein; at more negative values, the compounds are more affine for the protein. As seen in Figure 7, the control compounds GRL0617 and 12C have similar affinity for PLpro, which correlates with the inhibitory activity measured in vitro. Compound D24 has less affinity for PLpro than the control compounds (GRL0617 and 12C), and compounds D28, D04, and D59 have a similar affinity as the controls. Compound D60 has a ΔG_bind_ ~6 kcal/mol lower than controls; in contrast, D99 and D06 have a ΔG_bind_ ~10 kcal/mol lower than controls. The calculations of ΔG_bind_ performed using MMGBSA have smaller errors (Appendix A) than the differences between compounds, supporting the observations. These differences are less evident in the ΔG_bin_ calculation performed using molecular docking (Figure 7). For the complexes analyzed, van der Waals interactions (ΔE_vdw_) generate the largest contribution to the total binding free energy (ΔG_bind_), whereas the electrostatic interaction (ΔE_ele_) is counterbalanced by the polar desolvation energy (ΔG_GB_) (Table 1).

## 3. Materials and Methods

Compound GRL0617 is a promising candidate PLpro inhibitor that has been evaluated by antiviral, structural, and mechanistic studies. Following this analysis, a structural similarity search (Tanimoto Index of 85%) was performed in the PubChem database with millions of molecules (Figure 1). Thanks to the restrictions imposed, 17,889 compounds were selected and studied by molecular docking to evaluate their affinity for the GRL0617 binding site in PLpro.

### 3.1. Molecular Docking

The selected 17,889 compounds were docked into the binding cavity (rigid docking) of GRL0617 to PLpro using AutodockGPU suite. To evaluate the potential of the selected compounds, compounds with in vitro antiviral activity (GRL0617 and C12) were included in the docking. For docking, protein coordinates (PDB: 7CMD) with a resolution of 2.59 Å were used and were modified with Schrödinger’s protein preparation wizard to complete the structure (residues 220 to 231). Hydrogen atoms were added, charges were assigned, and the GRL0617 binding region was delimited by restricting it to a 20 Å cube centered at the center of mass of GRL0617 bound to PLpro. Overall, grid maps were calculated using the AutoGrid 4.0 option and the volume chosen for the grid maps was composed of 60 × 60 × 60 points, with a grid point spacing of 0.375 Å. To define the rotary bond in the ligand, the default option in the software was used. In the Lamarck genetic algorithm (LGA) couplings, an initial population of random individuals with a population size of 150 individuals, a maximum number of energy evaluations of 2.5 × 107, a maximum generation number of 27,000, a mutation rate of 0.02, and a crossover rate of 0.80 were employed. Each complex was constructed using the lowest coupled energy binding positions. van der Waals interactions were calculated using a smoothed 12–6 Lennard Jones potential, while hydrogen bonding interactions were evaluated using a 12–10 function incorporating a directionality term. That is, interactions deviating from ideal hydrogen bond geometries were progressively reduced.

The partial charges of each ligand were determined with the semi-empirical PM6-D3H4 method [23,24] implemented in MOPAC2016 software [25]. PM6-D3H4 introduces scattering and hydrogen bonding corrections to the PM6 method. The 3D representations of the docking results were analyzed using the VMD molecular graphics system.

From the molecular docking study, 500 molecules with binding energies between −14.1 and −11 kcal/mol were selected. The binding free energy obtained from the couplings was used to calculate the ligand efficiency and binding constant of each compound to PLpro.

To select compounds with possible inhibitory activity against PLpro and with potential to be used as a drug, the following computational estimates were considered: ligand efficiency (LE), physicochemical descriptors (molecular hydrogen bond acceptor, hydrogen bond donor, weight, topological polar surface area, rotational bond count, octanol/water partition coefficient, and molar refractivity) and estimates of drug properties (absorption, distribution, metabolism, excretion, and toxicological properties) [26]. Compounds with the highest ligand efficiency were selected, with a binding efficiency index (BEI) between 20 and 27 and a lipophilic ligand efficiency (LLE) between 5 and 7 [27].

### 3.2. Ligand Efficiency (LE)

LE allows the estimation of the binding affinity of a ligand to a target protein, weighted by the size of the molecule. LE (Equation (1)) is calculated from the dissociation constant (Kd) and the number of atoms other than H (HAC). Kd is obtained by Equation (2), where Δ*G*^0^ corresponds to the binding energy (kcal·mol^−1^) obtained from docking experiments, R the Renault constant (1.987207 cal·mol^−1^K^−1^), and T the temperature (298.15 K).
(1)$$LE=-\frac{2.303RT}{HAC}log\;log\;\left(K_{d}\right)$$
(2)$$K_{d}=10^{\frac{\Delta G^{0}}{2.303RT}}$$

### 3.3. BEI and LLE

These metrics are calculated based on the Kd obtained from molecular docking. BEI allows the estimation of the binding capacity weighted by the molar mass (Equation (3)), whereas LLE (Equation (4)) estimates the binding capacity with respect to its lipophilicity (clogP obtained from SwissADME webserver) [28,29].
(3)$$BEI=\frac{-log\left(K_{d}\right)}{MW}$$
(4)$$LLE=-log\left(K_{d}\right)-clogP$$

### 3.4. ADME-Tox Properties

The ADMET properties were calculated to estimate the drug viability of the 7 selected compounds. ADME-Tox profiles, which provide a preliminary prediction of the in vivo pharmacological behavior of the compounds, were obtained by this calculation. In addition, physicochemical properties such as molecular hydrogen bond acceptor (HBA), hydrogen bond donor (HBD), weight (MW), topological polar surface area (TPSA), rotational bond count (RB), octanol/water partition coefficient (LogP), and molar refractivity (MR) were calculated using the web server SwissADME [29]. The toxicological properties of the compounds were analyzed considering the toxicity rules of Lipinski, Ghose, Veber, and Pfizer [30].

### 3.5. Molecular Dynamics (MD) Simulation

MD simulations were performed with the seven selected couplings and two control compounds (GRL0617 and C12). Each model was constructed from the PLpro/ligand complex obtained from the molecular docking calculation in an explicit water-box model TIP3P. The protonation states of the ionizable residues at pH 7.0 were established with the H++ web interface based on calculations of the pK values of the ionizable groups [31]. The protein was parameterized with the ff19SB force field [32]. The domain that binds the Zn atom to PLpro (Cysteine residues 191, 194, 226, and 228) was parameterized with the ZAFF force field that is specific for proteins with metallic scepters of this atom [33]. The 7 selected molecules were parameterized with GAFF Force Field using the Antechamber module of AmberTools18, and the RESP charges were recalculated with a B3LYP/6-31G* level of theory in the TeraChem-GPU program [34].

MD simulations were performed with AMBER20-GPU using the following MD protocol [19,35]: (i) minimization and structural relaxation of water molecules with 2000 minimization steps and MD simulation with an NPT assembly (300 K) for 1000 ps using harmonic constraints of 10 kcal molÅ^−2^ for proteins and ligands; (ii) full structure minimization considering 6500 steps of conjugate gradient minimization; (iii) the minimized systems were progressively heated up to 300 K, with harmonic constraints of 10 kcal molÅ^−2^ for carbonate skeleton and ligand for 0.5 ns; (iv) the system was then equilibrated for 0.5 ns maintaining the constraints and then for 5 ns without constraints at 300 K in a canonical assembly (NVT); (v) finally, the total simulation duration was 200 ns for each system. During the MD simulations, the equations of motion were integrated with a time step of 2 fs in the NPT assembly at a pressure of 1 atm. The SHAKE algorithm was applied to all hydrogen atoms, and the van der Waals limit was set to 12 Å. The temperature was maintained at 300 K, employing the Langevin thermostat method with a relaxation time of 1 ps. The Berendsen barostat was used to control the pressure at 1 atm. Long-range electrostatic forces were accounted for using the particle-mesh Ewald (PME) approach. Data were collected every 1 ps during the MD tests [36]. Molecular visualization of the systems and MD trajectory analysis were carried out with the VMD software package.

### 3.6. Cluster Analysis

This statistical methodology separates the data points into several groups that exhibit similar properties and differ from the other groups. To perform the clustering, the density-based spatial method of applications with noise (DBSCAN) implemented in the CPPTRAJ tool was used [37]. This algorithm performs the separation by considering a cluster in the data space as a contiguous region of high point density, separated from other similar clusters by contiguous regions of low point density. Each analysis was performed with a cutoff distance of 1.5 Å based on the Root-Mean-Square Deviation (RMSD) of the ligand’s distinct hydrogen atoms and 5 points as a minimum for each cluster. According to this calculation, the representative structures of each simulation were obtained in relation to the position of the ligand in the protein.

### 3.7. Free Energy Calculation

The free energy of binding of each ligand in the last 100 ns of molecular dynamics simulations was estimated using “generalized Born and surface area continuum solvation molecular dynamics” MM-GBSA [38]. The analysis was performed on three subsets: the protein, the ligand, and the complex (protein–ligand). For each of these subsets, the total free energy (Δ*G_tot_*) was calculated as follows:(5)$$\Delta G_{tot}=E_{MM}+G_{solv}-T\Delta S_{conf}$$
where *E_MM_* is the bonded (bond, angle, and dihedral) and unbonded (electrostatic and Lenard-Jones) terms; *G_solv_* is the polar contribution of the solvation energy and the nonpolar contribution to the solvation energy; *T* is the temperature; and Δ*S_conf_* corresponds to the conformational entropy. Both *E_MM_* and *G_solv_* were calculated using AMBER20-GPU software with the generalized Born implicit solvent model. Δ*G_tot_* was calculated as a linear function of the solvent-accessible surface area, which was calculated with a probe radius of 1.4 Å. The binding free energy of SARS-CoV-2 PLpro and ligand complexes (Δ*G_bind_*) were calculated by the difference where the *G_tot_* values are the simulation averages.
(6)$$\Delta G_{bind}=G_{tot}\left(complex\right)-G_{tot}\left(protein\right)-G_{tot}\left(ligand\right)$$


### 3.8. Non-Covalent Interactions

The non-covalent interaction index (NCI) was calculated for each representative conformation (200 ns simulation) obtained from the cluster analysis. Non-covalent interactions, such as hydrogen bonds, steric repulsion, and van der Waals interactions, were identified and mapped using promolecular densities ($\rho^{pro}$), calculated as the sum of all atomic contributions. The NCI is based on the electron density (ρ), its derivatives and the reduced density gradients (s). The reduced density gradient is given by:(7)$$s=\frac{1}{2{\left(3\pi^{2}\right)}^{1/3}}\frac{\nabla \rho }{\rho^{4/3}}$$

These interactions are local and manifest in real space as low-gradient isosurfaces with low densities that are interpreted and colored according to the corresponding values of the sign $\left(\lambda_{2}\right)\rho$. The surfaces are colored on a blue-green-red scale according to the strength and type of interaction. Blue indicates strong, attractive interactions, green indicates weak van der Waals interactions, and red indicates strong unbound superposition. All calculations were performed with NCIPlot software [39].

## 4. Conclusions

A massive search for compounds structurally similar to GRL0617 was performed in the PubChem database. The affinity for the GRL0617 binding site of the ~18,000 compounds with a Tanimoto index of 85% was calculated by molecular docking, obtaining 500 with a higher affinity than GRL0617. Then, according to the physicochemical descriptors and ADME-Tox properties, the highest affinity compounds with feasibility to be used as drugs were selected. Using this methodology, out of ~109 million compounds from the PubChem database, 7 compounds were chosen as candidate PLpro inhibitors of SARS-CoV-2. The binding of each compound to the GRL0617 binding site to PLpro was evaluated by MD simulation and the protein–ligand interaction by NCI calculations. The constructed models were structurally stable with respect to protein structure, PLpro/ligand complex formation, and ΔG_bind_ calculated by MMGBSA. According to the ΔG_bind_, the controls, GRL0617 and 12C, may have similar binding affinities, which is consistent with experimental data. The affinity ΔG_bind_ by MMGBSA discriminates between the selected compounds and the controls. Although D24 shows a lower binding affinity than the controls, this compound binds to another region of the protein. Compounds D28, D04, and D59 could show a similar affinity for the binding site of GRL0617 to PLpro as the controls. Compounds D60, D99, and D06 present the highest affinity for PLpro, showing strong interactions with residues present in the GRL0617 binding site. Despite performing a comparative analysis with empirically tested molecules, being an in silico study, it is necessary to test the compounds by experimental assays. Considering this limitation and based on the analyses performed, we propose the seven compounds selected for in vitro studies to evaluate their inhibitory capacity.

## Figures and Tables

**Figure 1 pharmaceuticals-15-00986-f001:**
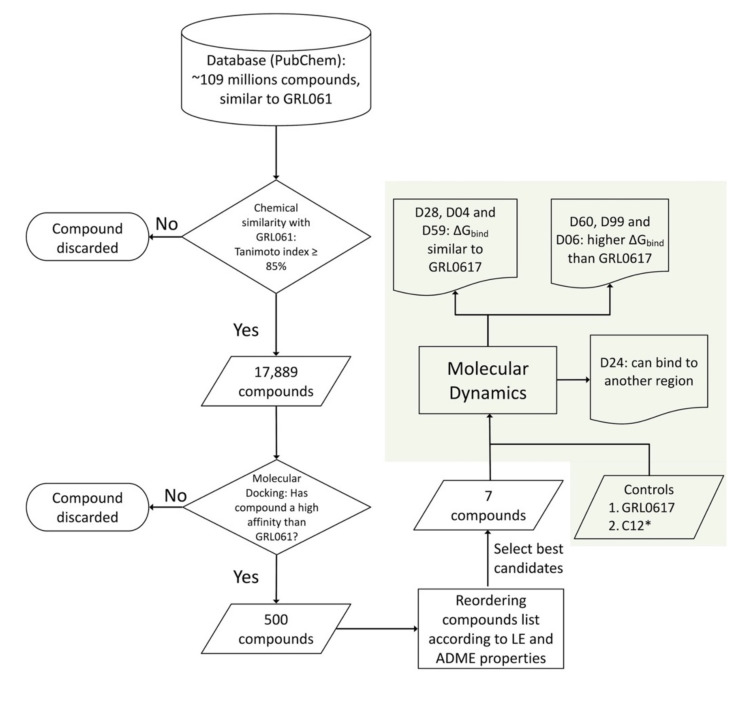
Diagram of the selection and evaluation of PLpro inhibitor compounds. Compound selection and evaluation process (green background). * PLpro inhibitor, in vitro tested, designed by chemical modification of GRL0617 [17].

**Figure 2 pharmaceuticals-15-00986-f002:**
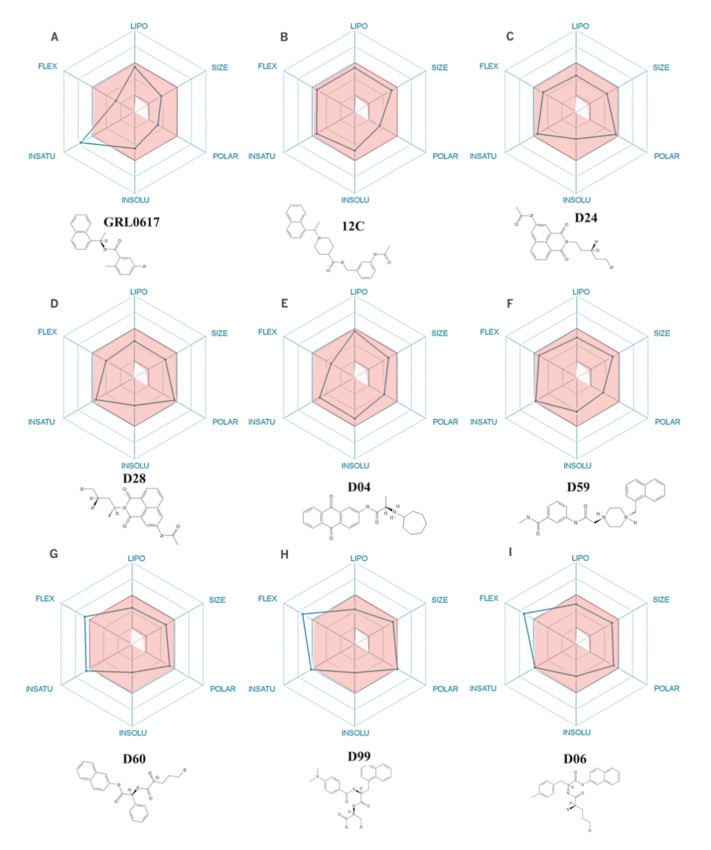
(**A**–**I**) Bioavailability radar of the selected compounds. The red area represents the optimal range for each property (lipophilicity: XLOGP3 between −0.7 and +5.0, size: MW between 150 and 500 g/mol, polarity: TPSA between 20 and 130 Å^2^, solubility: log S not greater than 6, saturation: fraction of carbons in sp3 hybridization, not less than 0.25, and flexibility: no more than 9 rotatable bonds). Under each bioavailability radar, the 2D chemical structure of each compound is observed.

**Figure 3 pharmaceuticals-15-00986-f003:**
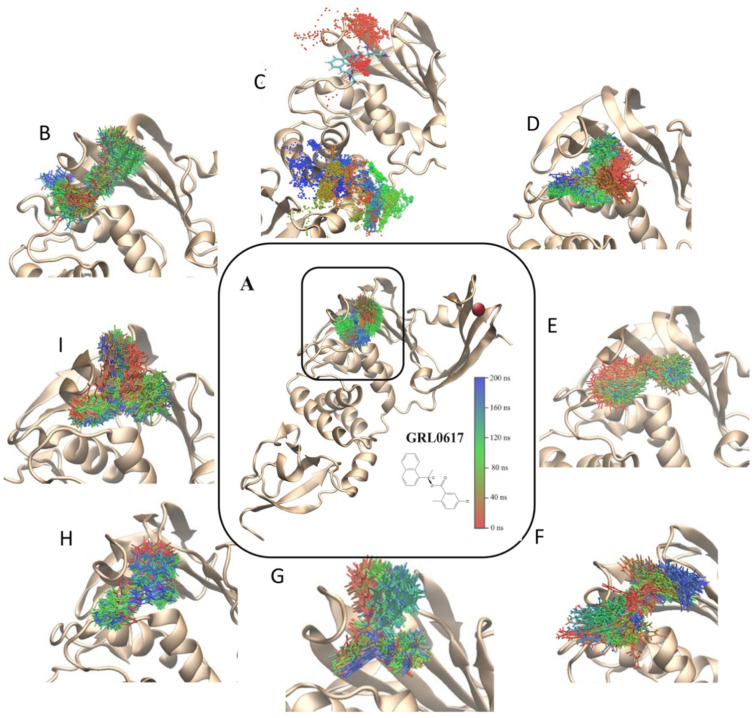
Conformations of the PLpro/ligand complex obtained by MD simulation. Crystallographic models of PLpro inhibitors GRL0617 and 12C (**A** and **B**, respectively) and seven new inhibitor compounds (D24, D28, D04, D59, D60, D99, and D06) obtained by in the in silico search and molecular docking (**C**–**I**) were used to perform 200 ns MD simulations (**A**,**B**,**D**–**I**). For compounds C12 (**B**), D24 (**C**), D28 (**D**), D04 (**E**), D59 (**F**), D60 (**G**), D99 (**H**), and D06 (**I**), an increase in the binding region of each ligand is observed. The red sphere corresponds to the Zn^2+^ ion. The position of the compounds is observed every ns (red 0 ns and blue 200 ns) of the simulation after superimposing the polypeptide chains. For compound D24 (**C**), 400 ns of simulation are observed (red 0 ns and blue 400 ns). The structural formula of each compound is shown in the Appendix A.

**Figure 4 pharmaceuticals-15-00986-f004:**
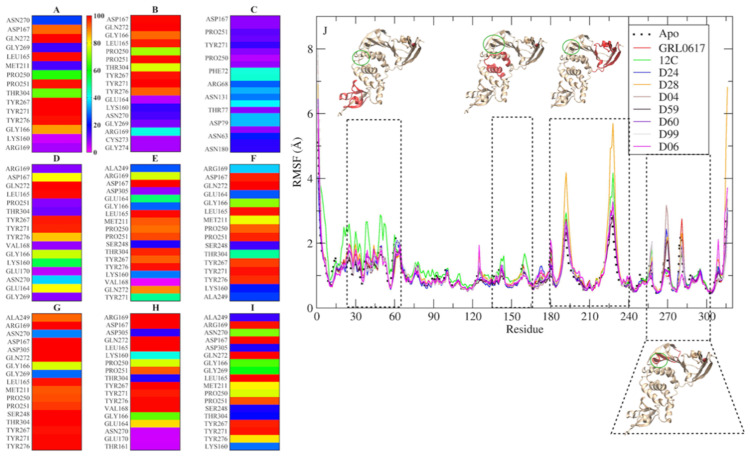
Residues of less than 3 Å (**A**–**I**) of the ligands and the perturbation they generate in the protein structure (**J**) during MD simulation. Simulations of 200 ns of GRL0617, 12C, D28, D04, D59, D60, D99, and D06 (**A**, **B**, **D**, **E**, **F**, **G**, and **I**, respectively) and 400 ns for D24 (**C**) were analyzed, showing the percentage of the simulation in which each ligand (blue 20% to red 100%) was found within 3 Å of the ligand. The perturbation of the protein structure (RMSF) was analyzed during the last 150 ns of each simulation and compared to the protein without ligand or Apo (black dots). In the figure, the region of greatest perturbation is marked (rectangle with a dotted line), and the region affected by the perturbation (red) and the ligand binding site (green circle) are highlighted on the protein.

**Figure 5 pharmaceuticals-15-00986-f005:**
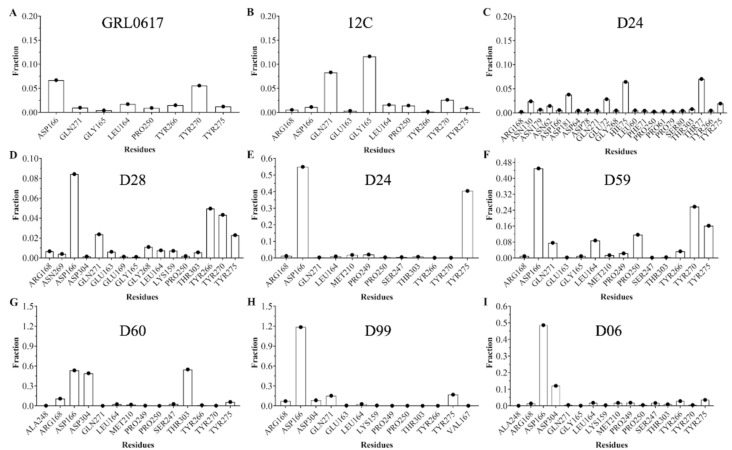
Fraction of intermolecular hydrogen bonds for SARS-CoV-2 PLpro interacting with controls (**A**,**B**) and selected ligands (**C**–**I**). The bar graph shows the most common hydrogen bonds formed between the pocket residues and the studied molecules. Values obtained from the CPPTRAJ script in AMBER.

**Figure 6 pharmaceuticals-15-00986-f006:**
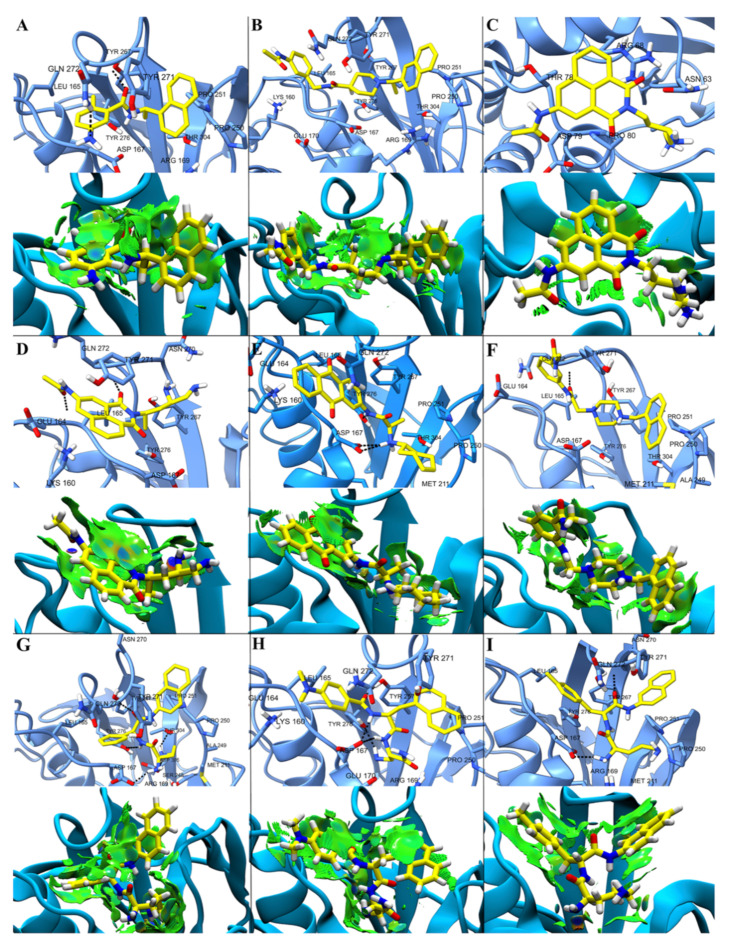
Non-covalent interactions in the representative conformation of the PLpro/ligand complex obtained by cluster analysis. The amino acids surrounding with controls GRL0617 and 12C (**A** and **B**, respectively) and selected ligands D24, D28, D04, D59, D60, D99, and D06 (**C**, **D**, **E**, **F**, **G**, **H**, and **I,** respectively) in the PLpro binding pocket are highlighted (upper figures), and in the two-dimensional PLpro/ligand interaction map (lower figures) the NCIPLOT isosurface gradient (0.5 au) is highlighted. Dashed lines indicate possible interactions between amino acids and adjacent ligands.

**Figure 7 pharmaceuticals-15-00986-f007:**
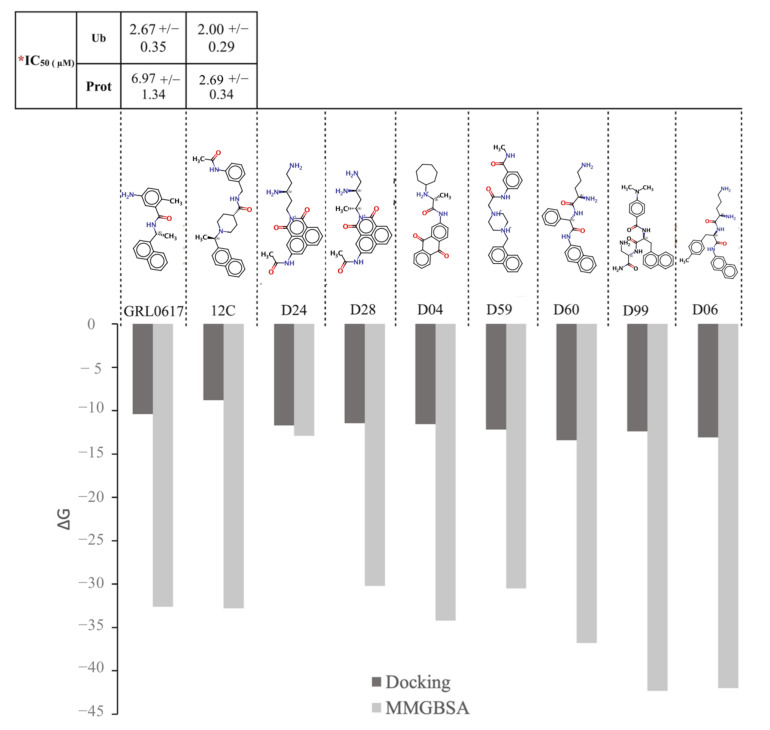
Free energy of binding (ΔG_bind_) of controls (GRL0617 and 12C) and selected compounds (D24, D28, D04, D59, D60, D99, and D06) to SARS-CoV-2 PLpro. ΔG_bind_ was calculated by molecular docking (black bar) and MMGBSA (gray bar) for each compound. MMGBSA calculations were performed from selected structures every 10 ns over the last 100 ns of the simulation. For each ligand, its chemical structure is highlighted, and for the controls, the IC50 is reported in the literature. * IC_50_ obtained from [17].

**Table 1 pharmaceuticals-15-00986-t001:** Predicted binding free energies (kcal/mol) and individual energy terms calculated from molecular dynamics simulation following the MM-GBSA protocol for PLpro complexes.

	Calculated Free Energy of Decomposition (kcal/mol)				
	ΔG*_bind_*	ΔE*_vdW_*	ΔE*_elect_*	ΔG*_gas_*	ΔG*_solv_*
GRL0617	−32.6	−37.9	−21.6	−59.52	26.9
12C	−32.8	−38.9	−22.1	−61.0	28.2
D24	−13.6	−22.2	−10.3	−32.5	18.9
D28	−30.2	−38.2	−31.9	−70.1	39.8
D04	−34.2	−42.4	−37,1	−79.5	45.3
D59	−30.5	−42.2	−25.5	−67.7	37.2
D06	−36.8	−40.3	−36.9	−77.2	40.5
D60	−42.0	−44.7	−42.2	−86.9	44.9
D99	−42.3	−44.8	−42.6	−87.4	45.1

## Data Availability

The data presented in this study are available in article or Appendix A.

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
