# Peer review of "Search for Novel Potent Inhibitors of the SARS-CoV-2 Papain-like Enzyme: A Computational Biochemistry Approach"

_pharmaceuticals, 2022, doi:10.3390/ph15080986_

Round 1

Reviewer 1 Report

In the manuscript “pharmaceuticals-1827961-peer-review-v1.pdf” (Search for novel potent inhibitors of the SARS-CoV-2 papain like enzyme: A computational Biochemistry approach), the authors propose seven candidates, PLpro inhibitors to initiate experimental studies that could establish their inhibitory capacity for use in future pharmacological treatments in the fight against COVID-19.

The paper completes the long list of manuscripts published in the last two years with a focus on the in silico searches of active compounds capable to combat the COVID-19 virus.  This work can certainly make an important contribution to these efforts. Furthermore, in the case of positive results from clinical trials, the identified compounds may pave the way for new antiviral agents capable to fight the disease.

In the referee's opinion, the manuscript covers the journal standards, and considering the potential impact of the paper results in the COVID-19 era, the manuscript could be accepted for publication in the Pharmaceuticals.

 The article is very clear and well organized. The computational protocol (similarity, molecular docking, MM-GBSA, ADME, and molecular dynamic simulations) is a standard one, but the software applied to achieve their goals are adequate, very well structured, presented, and actual. The authors were inspired to employ one of the most attractive and precise methods, molecular dynamic simulations, that could provide useful insights for molecular drug design targeting papain-like protease.

 Suggestions:

- Please reorganize Figure 3 to make it clear and easy to understand. There are too many and small pictures combined in one figure. By keeping this version, the main essence is completely lost. If the authors find any other appropriate solution, they are encouraged to provide it.

- In Figure 7, the 2D structures of the compounds must be rotated in the drawing program

Conclusion section - Some words should be spent to briefly highlights limits still existing in computational-only performed drug research 

Author Response

Reviewer 1: Thank you for your comments and suggestions. To respond in detail, we made the following changes:

Suggestion 1. Please reorganize Figure 3 to make it clear and easy to understand. There are too many and small pictures combined in one figure. By keeping this version, the main essence is completely lost. If the authors find any other appropriate solution, they are encouraged to provide it.

Response: Figure 3 was modified to highlight the position of each ligand at the binding site, reducing the redundant information of the rest of the protein not involved in binding.

Suggestion 2. In Figure 7, the 2D structures of the compounds must be rotated in the drawing program

Answer: The 2D structures were changed to ones drawn in the MarvinSketch program.

Suggestion 3. Conclusion section - Some words should be spent to briefly highlights limits still existing in computational-only performed drug research.

Response: The last lines (451 to 455) of the conclusion were modified to indicate the limitation of in silico studies "Despite performing a comparative analysis with empirically tested molecules, being an in silico study it is necessary to test the compounds by experimental assays. Considering this limitation and based on the analyses performed, we propose the 7 compounds selected for in vitro studies to evaluate their inhibitory capacity."

Reviewer 2 Report

Pharmaceuticals-1827961-peer-review-v1

This manuscript describes an in silico study for the discovery of potential PLpro inhibitors. The authors started with a ligand-based approach to screen millions of compounds in the PubChem database, followed by molecular docking of the resulting ~18K compounds. Of the 500 compounds with better docking scores than the positive control compound GRL061, 7 were selected as they had the best ligand efficiencies (docking scores over number of heavy atoms) and possibly good ADMET properties? Finally, 6 out of the 7 compounds, together with 2 positive controls, showed good binding to the protease domain of PLpro in MD simulations.

The authors are praised for using some of the best computational chemistry programs for calculations and visualization, including the choice of MD engine AMBER20-GPU, the derivation of RESP charges for small molecules, and visualization by VMD.

However, some minor issues should be addressed.

First, the authors are encouraged to add a negative control (e.g. a docked compound with very low docking score), in addition to the two positive controls, for MD simulations to further improve the rigor of their computational results. This may not be entirely necessary considering compound D24 did egress out of its initial docked site and moved towards binding another site. But the authors should at least discuss the possibility that any docked compound could show good/stable binding to the protein in their MD simulations.

Second, where did compound 12C come from and what its activity was in wet experiments? Its first appearance was Line 126. If it’s from Ref 17, its protein IC50 should be 2.69 uM instead of 6.69 (Figure 7).

Third, the authors should clarify how the 7 compounds were selected. It’s hard to tell if they are only ranked by LE or if ADMET properties are also considered.

Fourth, “Supplementary Materials” and “Author Contributions” are missing.

Author Response

Reviewer 2: Thank you for your comments and suggestions. To respond in detail, we made the following changes:

Comment 1: First, the authors are encouraged to add a negative control (e.g. a docked compound with very low docking score), in addition to the two positive controls, for MD simulations to further improve the rigor of their computational results. This may not be entirely necessary considering compound D24 did egress out of its initial docked site and moved towards binding another site. But the authors should at least discuss the possibility that any docked compound could show good/stable binding to the protein in their MD simulations.

Answer: As you indicate, non-specific interaction is inherent of Protein/ligand binding, more so in simulations. However, this non-specific interaction is maintained for short periods of simulation time, occupying sites randomly. Thus, as observed with compound D24, the interaction with the binding site is lost in less than 50 ns and the compound binds to another region where it is maintained for about 100 ns. To highlight this phenomenon, a comment was added to line 178 "The simulation time is sufficient to discriminate between ligands that can bind specifically to the GRL0617 binding site and those that cannot (D24). However, for the case of D24, with a 400 ns simulation we could observe a binding site distinct from that of GRL0617 and maintained for more than 200 ns. The stability shown could indicate a new binding region that could be explored in future work."

Comment 2. Second, where did compound 12C come from and what its activity was in wet experiments? Its first appearance was Line 126. If it’s from Ref 17, its protein IC50 should be 2.69 uM instead of 6.69 (Figure 7).

Answer: To clarify the origin of compound 12C, the following line (131 to 132) was incorporated in 2.1 Molecular Docking "GRL0617 is one of the first PLpro-specific inhibitors and 12C was designed by incorporating an N-phenylacetamide group in the amide group to GRL0617[17]". In addition, the IC50 value of compound 12C was modified (from 6.69 to 2.69) in Figure 7 to correct the error.

Comment 3. Third, the authors should clarify how the 7 compounds were selected. It’s hard to tell if they are only ranked by LE or if ADMET properties are also considered.

Answer: The first paragraph (line 147 to 161) of item 2.2. (Physicochemical Descriptors and ADME Properties) was reorganized to clarify the selection process “the 20 with the highest value of this physicochemical descriptor. To assess the drug viability of each compound, six physicochemical properties were calculated: lipophilicity, size, polarity, solubility, flexibility, and saturation. After discarding compounds that did not meet more than 2 pharmacological criteria, the 7 with the highest LE were selected. As shown in Figure 2, compounds D24, D28, D04, D59, D60, D60, D99, D06, 12C and GRL0617 show up to 2 parameters out of range. Like GRL0617, which exceeds one ADEM-Tox property (unsaturation), 3 selected compounds exceed one parameter: compound D06 exceeds the flexibility range and compounds D99 and D60 exceed the flexibility and unsaturation parameter. As shown in Figure S4, all compounds, including the controls GRL0617 and 12C, show a high probability of gastrointestinal absorption, while the controls also show a high probability of brain penetration. Looking at the chemical structures of these compounds (Figure S3), it is identified that molecules D24, D28, D59, D60, D99 and D06 present a naphthyl group, as do GRL0617 and 12C, while compound D04 presents a (9,10-dioxoanthracene-2-yl) amino]-1-oxopropan-2-yl group”. In addition, the numbering of Supplementary Figures S3 and S4 was changed to match the text.

Comment 4. Fourth, the "Supplementary Materials" and "Author Contributions" are missing.

Answer: Each author's contribution has been added and Supplementary Materials is attached for your review.